# Heterogeneous Aligned Fusion for Survival Classification with Missing Modalities

**Zheng Zheng**[1] 🆔                                   ZXZ7934@MAVS.UTA.EDU
**Yuzhi Guo**[1]                                        YUZHI.GUO@MAVS.UTA.EDU
**Xiao Hu**[1]                                          XXH3416@MAVS.UTA.EDU
**Yuwei Miao**[1]                                       YXM9326@MAVS.UTA.EDU
**Hehuan Ma**[1]                                        HEHUAN.MA@MAVS.UTA.EDU
**Jean Gao**[1]                                         GAO@UTA.EDU
**Junzhou Huang**[1]                                    JZHUANG@UTA.EDU
[1] *University of Texas at Arlington, Arlington, TX, USA*

**Editors:** Accepted for publication at MIDL 2026

## Abstract

Accurate survival classification is essential for guiding personalized treatment in head and neck cancer. Heterogeneous biomedical data, from histopathology to clinical and laboratory measurements, offer complementary prognostic value but differ in dimensionality, reside in incompatible feature spaces, and are frequently missing, making robust multimodal learning challenging. To address this, we propose **HAF (Heterogeneous Aligned Fusion)**, a three-stage framework for survival classification under heterogeneous and incomplete multimodal inputs. HAF (i) uses detachment and prognostic supervision to obtain stable representations, (ii) performs lightweight global alignment that projects all modalities into a shared latent space while preserving patient-level discriminability, and (iii) enforces monotonic robust fusion that encourages performance to remain stable or improve when modalities are added. To the best of our knowledge, HAF is the first approach that jointly leverages all seven modalities in the HANCOCK cohort. Extensive comparisons against representative late-, early-, attention-based, and bilinear-interaction fusion methods demonstrate that HAF consistently improves both accuracy and robustness under heterogeneous and partially missing modalities. Codes are released at https://github.com/zz9tf/HAF.git.
**Keywords:** Heterogeneous Aligned Fusion (HAF), multimodal learning, head and neck cancer, survival classification, pathology imaging, MIL

## 1. Introduction

Accurate survival classification is central to precision oncology, enabling risk-adaptive decision making for patients with head and neck squamous cell carcinoma (HNSCC) (Tian et al., 2025). Modern cohorts combine high-dimensional pathology imaging with heterogeneous clinical and laboratory measurements, which differ in dimensionality, reside in incompatible feature spaces, and are often missing (Wissel et al., 2023; Li et al., 2024; Khagi and Kwon, 2020; Li and Tang, 2024; Pan et al., 2020; Aly et al., 2023; Wu et al., 2024; Reza et al., 2024). This makes robust multimodal learning particularly challenging, as both representation compatibility and modality availability become fundamental bottlenecks.

A natural starting point for multimodal survival classification is to build strong unimodal predictors. Pathology-only systems such as CLAM-style MIL models (Tian et al., 2025)

provide reliable prognostic signals, yet treat each modality independently and therefore cannot exploit complementary clinical or laboratory information.

Fusion-based multimodal designs address this limitation but introduce new challenges. These designs span different abstraction levels, including early fusion (Jaegle et al., 2021) that concatenates heterogeneous features at the data level, late fusion (Tian et al., 2025) that aggregates modality-specific predictions, attention-based (Raza et al., 2025; Dang et al., 2024) fusion that models cross-modality dependencies, and bilinear interaction frameworks (Benkirane et al., 2025) that explicitly capture pairwise cross-modality interactions (Jaegle et al., 2021; Tian et al., 2025; Raza et al., 2025; Dang et al., 2024; Benkirane et al., 2025). However, they implicitly assume well-aligned modality embeddings. In reality, heterogeneous modalities remain misaligned (Li et al., 2024; Khagi and Kwon, 2020; Li and Tang, 2024), allowing low-quality channels to propagate misleading signals that degrade the fused representation.

A parallel line of work introduces explicit cross-modality alignment, e.g., contrastive learning or subspace matching (Radford et al., 2021; Cicchetti et al., 2024; Kamboj and Do, 2025). However, these methods typically assume complete modality availability and do not address missing-modality scenarios. Another line of work focuses on robustness under missing modalities via monotonicity loss to train (Aly et al., 2023; Wu et al., 2024; Reza et al., 2024; Li et al., 2025), but these approaches operate in unaligned feature spaces and ignore structured cross-modal relationships. In clinical practice, however, multimodal data are rarely complete or equally reliable. Different modalities may vary substantially in acquisition quality, availability, and diagnostic relevance across patients, making naive fusion fragile in real-world settings.

These limitations suggest that reliable multimodal survival classification requires: (i) robustness under missing modalities, (ii) aligned latent geometry across heterogeneous modalities, and (iii) monotonic behavior such that incorporating additional modalities does not degrade performance. To operationalize these observations, we propose HAF (Heterogeneous Aligned Fusion), a three-stage framework that jointly enforces cross-modality alignment and monotonic robust fusion. Specifically, HAF decomposes multimodal survival learning into three stages: (i) *decoupled representation learning*, which detaches the outputs between stages to decouple representation learning, alignment, and fusion objectives, stabilizing optimization and preventing cross-objective gradient interference; (ii) *global latent alignment*, which maps heterogeneous modalities into a shared low-rank patient space; and (iii) *monotonic robust fusion*, which enforces structured modality substitution under missing-modality conditions.

We evaluate HAF on the HANCOCK dataset (Dorrich et al., 2025), which provides an unprecedented multimodal setting combining whole-slide imaging (WSI), tissue microarrays (TMA), and five structured clinical descriptors, capturing complementary aspects of tumor biology. In this work, we formulate this survival classification as a binary classification problem following the official target definition of the HANCOCK cohort. In evaluation, HAF demonstrates stable optimization, improved cross-modality compatibility, and strong robustness under missing or noisy inputs, outperforming representative late-, early-, and attention-based baselines. In conclusion, HAF offers a principled view of multimodal interaction: unimodal semantics are stabilized, modalities are aligned into a substitutable shared geometry, and fusion remains reliable even under realistic missing-modality conditions.

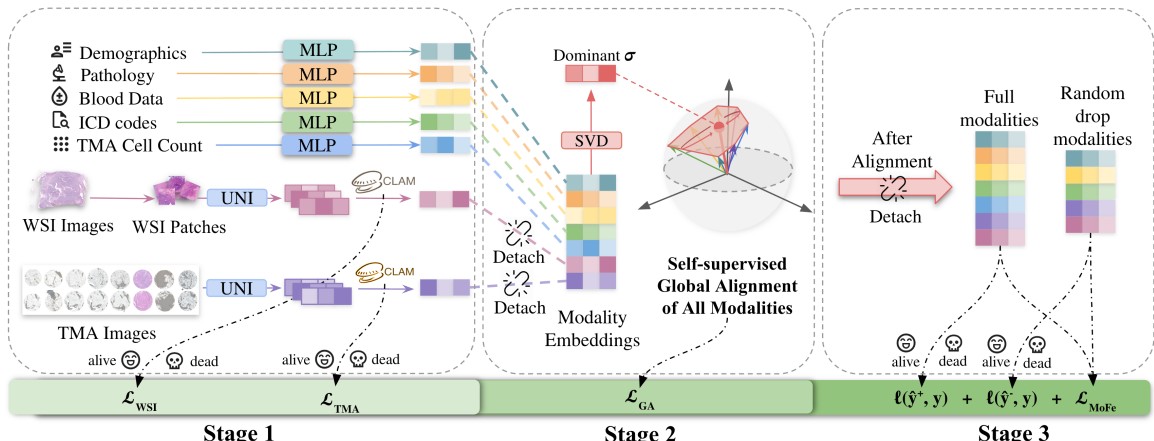

Figure 1: **Overview of the HAF framework.** Stage 1 learns stable unimodal embeddings under detached supervision. Stage 2 globally aligns modality embeddings into a shared space. Stage 3 fuses aligned features with monotonic-fusion training and a monotonic constraint, promoting robustness under missing modalities.

## 2. Related Work

### 2.1. Multimodal fusion paradigms for survival classification

Recent multimodal survival approaches can be broadly discussed under several representative fusion paradigms, including: (A) *Early fusion*, which concatenates heterogeneous modality representations into a single joint feature vector before prediction; (B) *Late fusion*, as in MDLM (Tian et al., 2025), which first produces modality-specific predictions and then aggregates these predictions at the decision level; (C) *Attention-based fusion*, where methods such as PS3 (Raza et al., 2025) perform cross-modal attention directly over modality representations to model feature-level dependencies. Within this family, MFMF (Dang et al., 2024) adopts a distinct asymmetric strategy, using auxiliary modalities to compute attention over a single target modality; (D) *Bilinear interaction frameworks*, exemplified by CustOmics (Benkirane et al., 2025), which explicitly model pairwise multiplicative interactions between modality representations.

### 2.2. Cross-modality alignment

Recent work has shown that aligning modalities before fusion can yield more semantically coherent representations, as demonstrated in CLIP-style contrastive models and cross-aligned fusion frameworks (Radford et al., 2021; Rajora et al., 2025; Zhao et al., 2025). Contrastive alignment, however, requires curated modality pairs and becomes inefficient as the number of modalities increases. Facing this limitation, approaches shift toward *geometry-driven alignment*, which aligns modalities through a shared latent structure without requiring paired supervision, typically via SVD-based formulations (Cicchetti et al., 2024; Kamboj and Do, 2025; Liu et al., 2026).

## 2.3. Missing-modality robustness

Missing data is pervasive in clinical workflows where imaging, assays, or laboratory tests may be absent for logistical or cost-related reasons (Pan et al., 2020; Aly et al., 2023; Wu et al., 2024; Reza et al., 2024). Generative imputation strategies (Hao et al., 2024; Liang et al., 2022; Qin et al., 2023) attempt to synthesize absent modalities but can introduce hallucinations or low-fidelity artifacts (Sun et al., 2024), making reliability difficult to guarantee. Random-modality training and stochastic gating (Li et al., 2025; Wei et al., 2025) provide an alternative by encouraging models to remain predictive even when some modalities are dropped during training.

## 3. Methods

Building on prior work in multimodal fusion, alignment, and robustness, we observe that existing approaches are not explicitly designed to handle systematic modality missingness, modality-quality variability, and the geometry of a shared alignment space in clinical settings. In addition, this work focuses on fusion strategies operating at the representation level, where each modality is first encoded into a fixed-dimensional embedding[1] before fusion. We select and reimplement baselines from each paradigm for representation-level fusion strategy comparison, including MDLM for late fusion, PS3 and MFMF for attention-based fusion, and CustOmics for bilinear interaction. This design is motivated by the substantial heterogeneity in raw data formats across modalities, which makes direct end-to-end fusion difficult to compare in a controlled manner.

### 3.1. Multimodal Inputs and Task Definition

We study binary survival status classification on the HANCOCK cohort (Dorrich et al., 2025), strictly following the official target definition provided in the dataset. Specifically, for survival classification, the class labels correspond to the patient survival status (*deceased* vs. *living*) at the time of the last available follow-up ($y \in \{0, 1\}$), consistent with the original HANCOCK setting; Patients with non–tumor-specific death are excluded, consistent with the original protocol. Each patient provides up to seven heterogeneous modalities: WSI and TMA histopathology, TMA-derived cell-density maps, clinical metadata, pathological staging, blood biomarkers, and ICD diagnostic codes. In practice, not all patients are fully observed; imaging and laboratory assays may be unavailable for logistical or cost-related reasons, leading to patient-specific subsets of observed modalities. Formally, for each patient $i$ we denote by $\mathcal{M}_i \subseteq \mathcal{M}$ the set of available modalities and by $\{x_i^{(m)}\}_{m \in \mathcal{M}_i}$ the corresponding inputs, and the goal is to learn a predictor $f$ that maps these heterogeneous, partially missing observations to a binary event classification: $\hat{y}_i = f(\{x_i^{(m)}\}_{m \in \mathcal{M}_i}), \hat{y}_i \in \{0, 1\}$. The preprocessing in this work addresses feature-level missingness within existing modalities, while modality-level missingness (entire modalities unavailable) is handled separately via random-modality masking in Stage 3. Additional dataset statistics and evaluation details are provided in the appendix.

---

1. "Fixed-dimensional" refers to a unified embedding space rather than frozen encoders; modality encoders remain trainable through lightweight adaptation modules.

## 3.2. Modality Representations

For WSI, slides are processed using the CLAM tile extraction pipeline, and tile-level embeddings are extracted with the UNI foundation model (Chen et al., 2024), followed by patient-level aggregation. For TMA, embeddings are extracted from the provided TMA PNG images using the same UNI model, rather than from the original TMA SVS files, and are likewise aggregated at the patient level. The remaining five modalities are compact tabular descriptors: $X^{(m)} \in \mathbb{R}^{d_m}, \quad m \in \mathcal{M}_{\text{tab}}, \quad d_m \leq 51$, where $\mathcal{M}_{\text{tab}} = \{$Cell, Clin, Path, Blood, ICD$\}$. All tabular features are preprocessed using min–max normalization, one-hot encoding, and imputation with the most frequent value, following the preprocessing protocol in the HANCOCK dataset. This yields a unified representation where high-dimensional imaging bags and low-dimensional tabular vectors coexist, but still differ substantially in scale, noise level, and semantic content, motivating the alignment and fusion strategy introduced in the following sections.

## 3.3. Stage 1: Prognostic Pathology Representation Learning

Each pathology modality (WSI or TMA) is represented as a variable-length bag of patch embeddings $H \in \mathbb{R}^{N \times 1024}$, where $N$ depends on sampled tissue content. We adopt the gated-attention MIL encoder from CLAM (Lu et al., 2021).

To obtain compact and expressive instance features, each 1024-dimensional patch embedding is first projected using a linear mapping $w_l \in \mathbb{R}^{1024 \times d_1}$. Two learnable projection matrices $U, V \in \mathbb{R}^{d_1 \times d_2}$ generate gated-attention features, and a learnable query vector $w_a \in \mathbb{R}^{d_2 \times 1}$ computes instance importance:

$$a = \text{softmax}\Big( \big[\tanh\big((Hw_l)V\big) \odot \sigma\big((Hw_l)U\big)\big] w_a \Big). \tag{1}$$

The slide-level representation for modality $m$ is obtained by weighted aggregation:

$$z^{(m)} = (Hw_l)^\top a, \tag{2}$$

where $m \in \{\text{WSI}, \text{TMA}\}$ and $d_2$ denotes the attention dimension. Each modality-specific embedding is supervised using a cross-entropy loss combined with instance-level regularization:

$$\mathcal{L}_{\text{img}}^{(m)} = \mathcal{L}_{\text{CE}}(y, \hat{y}^{(m)}) + \mathcal{L}_{\text{inst}}^{(m)}. \tag{3}$$

We use $d_1 = 64$ and $d_2 = 32$. The resulting pathology embeddings are detached before Stage 2 to preserve their prognostic semantics and avoid distortion during multimodal training.

## 3.4. Stage 2: Global Aligned Shared Latent Projection

This stage aligns modalities into a shared latent space, ensuring that downstream robustness operates on substitutable rather than incompatible modality embeddings. To achieve this, we first project each modality into a common latent space using small MLPs, then apply global alignment losses to encourage their convergence onto a unified consensus signal. This yields a coherent multimodal representation that amplifies high-quality modalities while suppressing noisy or weak ones.

### 3.4.1. LATENT PROJECTION INTO A SHARED SPACE

To place all modalities on a comparable footing, each modality-specific embedding $z^{(m)}$ is mapped into a shared latent space through a lightweight two-layer MLP:

$$u^{(m)} = \phi^{(m)}(z^{(m)}) \in \mathbb{R}^{d_{\text{out}}}, \quad m \in \mathcal{M}, \tag{4}$$

where $\mathcal{M} = \{\text{WSI}, \text{TMA}, \text{Clin}, \text{Path}, \text{Blood}, \text{ICD}, \text{Cell}\}$ and $d_{\text{out}} = 128$. The alignment treats all modalities symmetrically, avoiding vision-centric or tabular-centric bias.

### 3.4.2. GLOBAL ALIGNMENT VIA SINGULAR VALUE DECOMPOSITION

Singular value decomposition (SVD) decomposes a set of vectors into orthogonal directions ordered by how much signal the data concentrate along each direction. The largest singular value $\sigma_1$ corresponds to the dominant component shared across vectors, while smaller singular values capture weaker signals. In our setting, if modalities project toward a common disease-related direction, most information accumulates in this dominant component (large $\sigma_1$), and the remaining components become comparatively weak (small $\sigma_2, \ldots, \sigma_k$).

Following the principled alignment framework of Liu et al. (Liu et al., 2026), we leverage this property by encouraging all projected modality embeddings to concentrate their variation along the dominant component of the patient-specific matrix $U = [u^{(1)}, \ldots, u^{(M)}] \in \mathbb{R}^{d_{\text{out}} \times M}$. Formally, SVD decomposes $U$ into $U = Q\Sigma R^\top$, where $Q = [q_1, q_2, \ldots, q_k] \in \mathbb{R}^{d_{\text{out}} \times k}$ contains orthonormal left singular vectors, and $\Sigma = \text{diag}(\sigma_1, \sigma_2, \ldots, \sigma_k), k = \min(d_{\text{out}}, M)$ stores the singular values in descending order. Here, $\sigma_1$ quantifies the strength of the dominant shared component across modalities, and $q_1$ is the corresponding direction in the latent space. When $\sigma_1 \gg \sigma_2, \ldots, \sigma_k$, the decomposition becomes approximately rank-1, meaning that all modality embeddings align along $q_1$ and express a common latent signal. We adopt two complementary losses from Liu et al. (Liu et al., 2026) to enforce this behavior. First, the *singular emphasis loss* increases the dominance of $\sigma_1$, promoting alignment of all modalities toward the shared component:

$$\mathcal{L}_{\text{SV}} = -\log \frac{\exp(\sigma_1/\tau)}{\sum_{j=1}^{k} \exp(\sigma_j/\tau)}. \tag{5}$$

Second, during this alignment process, different patients could collapse to the same direction. To retain patient-level distinction, we adopt their *dominant-direction discriminability loss* that separates patients based on their dominant singular vectors:

$$\mathcal{L}_{\text{PD}} = -\frac{1}{B} \sum_{i=1}^{B} \log \frac{\exp((q_1^{(i)})^\top q_1^{(i)}/\tau)}{\sum_{j=1}^{B} \exp((q_1^{(i)})^\top q_1^{(j)}/\tau)}. \tag{6}$$

The total alignment objective is

$$\mathcal{L}_{\text{GA}} = \mathcal{L}_{\text{SV}} + \lambda_{\text{PD}}\mathcal{L}_{\text{PD}}. \tag{7}$$

In our experiments, we set the patient-discriminability weight to $\lambda_{PD} = 0.01$, which balances alignment strength and inter-patient separation.

### 3.5. Stage 3: Fusion with Robust Modality Collaboration

We perform fused multimodal classification using the aligned features from Stage 2. For each patient, let $\{u^{(m)}\}_{m\in\mathcal{M}}$ denote the aligned modality embeddings and $\mathcal{M}_{\text{obs}} \subseteq \mathcal{M}$ the set of modalities observed for that patient. We construct a fused representation by concatenating the available modalities:

$$h = \text{concat}\left(\{u^{(m)}\}_{m\in\mathcal{M}_{\text{obs}}}\right), \tag{8}$$

and obtain a scalar logit using a shared fusion MLP,

$$s = \phi(h), \qquad \hat{y} = \sigma(s), \tag{9}$$

where $\sigma(\cdot)$ is the sigmoid function.

**Random-modality Drop training.** To ensure robustness under missing or unreliable modalities, we adopt random-modality drop (Li et al., 2025) masking on top of the aligned geometry, enabling the model to rely on substitutable modalities rather than raw heterogeneous features. At each iteration, we randomly mask a subset of modalities to obtain a reduced-modality representation $h^-$ and its classification $\hat{y}^-$, alongside the full-modality classification $\hat{y}^+$. This encourages the model to rely on informative modalities while remaining stable when others are absent.

**Monotonic collaboration constraint.** To guarantee that incorporating more modalities never degrades performance, we impose a monotonicity loss:

$$\mathcal{L}_{\text{MoFe}} = \max\left(\ell(\hat{y}^+, y) - \ell(\hat{y}^-, y), 0\right), \tag{10}$$

which penalizes cases where the full-modality classification is worse than that of a reduced subset.

**Fusion objective.** The final fusion loss combines full-modality supervision, reduced-modality supervision, and the monotonicity constraint:

$$\mathcal{L}_{\text{fusion}} = \ell(\hat{y}^+, y) + \ell(\hat{y}^-, y) + \lambda_{\text{MoFe}}\,\mathcal{L}_{\text{MoFe}}, \tag{11}$$

where we set $\lambda_{\text{MoFe}} = 0.1$ in our experiments. The full objective therefore enforces consistency across complete and reduced-modality inputs, while the monotonicity term prevents classification reversals under modality removal. We found this combination to yield stable optimization and improved robustness in practice.

### 3.6. Overall Training Objective

The full HAF objective integrates the three stages through four loss blocks:

$$\mathcal{L}_{\text{total}} = \underbrace{\mathcal{L}_{\text{WSI}} + \mathcal{L}_{\text{TMA}}}_{\text{Stage 1}} + \underbrace{\mathcal{L}_{\text{GA}}}_{\text{Stage 2}} + \underbrace{\mathcal{L}_{\text{fusion}}}_{\text{Stage 3}}. \tag{12}$$

Here, $\mathcal{L}_{\text{WSI}}$ and $\mathcal{L}_{\text{TMA}}$ include $\mathcal{L}_{\text{CE}}$ and instance-level regularization terms as defined in Eq. (3). The global alignment loss $\mathcal{L}_{\text{GA}}$ follows Eq. (7). The fusion objective $\mathcal{L}_{\text{fusion}}$

follows Eq. (11). Crucially, the outputs of each stage are detached before being passed to the next stage. This decoupling prevents conflicting gradients from dominating weaker modalities and ensures that each learning objective, including (i) pathology semantics, (ii) cross-modality compatibility, and (iii) robustness to missing data, is optimized without being overridden by others.

## 4. Experiments

### 4.1. Experimental Setup

We adopt the same modality representations and preprocessing pipeline described in Sec. 3, including CLAM+UNI for histopathology and the official HANCOCK protocol for structured modalities. We applied a strict patient-level 10-fold random cross-validation protocol across all methods, and reported accuracy and AUC as the primary evaluation metrics. All metrics are computed exclusively at the patient level to avoid data leakage across splits. All models are trained under the same optimization protocol without method-specific hyperparameter tuning. We apply early stopping and select the best model based on validation AUC (maximum 200 epochs, patience 25). A `ReduceLROnPlateau` scheduler is used with patience 15 and factor 0.5. Unless otherwise stated, all experiments are conducted on an NVIDIA RTX A6000 GPU; full 10-fold training for HAF requires roughly 4 hours in total.

### 4.2. Baselines and Variants

We evaluate HAF against a comprehensive set of baselines designed to reflect four levels of comparison: (i) pathology-only references, (ii) fusion models and HAF variants, (iii) parameter-free aggregation baselines, and (iv) representative multimodal methods from the literature.

**Pathology-only baselines.** The official HANCOCK benchmarks consist of **WSI-CLAM**, **TMA-CLAM**, and **WSI+TMA-CLAM**, which serve as unimodal and dual-modality pathology references. We additionally reproduce WSI+TMA-CLAM using the released UNI features to ensure evaluation consistency. We report the original HANCOCK results only for contextual reference, as their evaluation protocol is defined at the slide level, whereas all experiments in our work adopt a strict patient-level split.

**Naive fusion models.** Beyond pathology-only baselines, we include two naive fusion models: a **WSI+TMA Fusion MLP** and an **All-Modality Fusion MLP**, both of which directly concatenate modality embeddings without any alignment or robustness mechanisms. These baselines quantify the limitations of simple multimodal aggregation in heterogeneous feature spaces.

**Alignment and robustness variants.** To examine the effects of HAF's core components, we further evaluate **Global Alignment (GA)**, the SVD-based latent alignment from Stage 2; **CLIP Alignment**, a vision-centric contrastive baseline anchoring non-visual modalities to WSI; and **Random-Modality Drop**, the robustness mechanism from Stage 3. We also test their combinations to assess whether alignment and robustness act synergistically.

**Parameter-free aggregation baselines.** Following recent work on simple multimodal interaction (Zhang et al., 2023), we additionally evaluate parameter-free pooling strategies

on the aligned modality embeddings, including mean, sum, and max pooling. These baselines test whether robustness gains can be achieved by straightforward aggregation alone, without any learnable fusion mechanism.

**Comparable multimodal methods.** Finally, we compare against representative multimodal fusion approaches, including **PS3**, **MDLM**, **MFMF**, and a **Bilinear Interaction** model, to situate HAF relative to existing fusion strategies. All literature baselines are re-implemented as representation-level fusion modules operating on fixed-dimensional modality embeddings, following the original methodological descriptions. All methods use identical encoder architectures and the same preprocessing and training protocol, and are compared under the same setting.

Table 1: **Overall quantitative results on the HANCOCK dataset.**

| Method | accuracy (mean ± std) | AUC (mean ± std) |
|---|---|---|
| *Pathology-only baselines* | | |
| WSI-CLAM | – | 0.65 |
| TMA-CLAM | – | 0.52 |
| WSI+TMA-CLAM | – | 0.69 |
| WSI+TMA-CLAM (reproduced) | 0.712±0.087 | 0.679±0.119 |
| *Fusion models and HAF variants* | | |
| WSI+TMA Fusion MLP | 0.739±0.041 | 0.668±0.133 |
| All-Modality Fusion MLP | 0.748±0.046 | 0.694±0.113 |
| Global Alignment (GA) | **0.752±0.047** | 0.698±0.127 |
| CLIP Alignment | 0.741±0.074 | 0.697±0.103 |
| Random-Modality Drop | 0.748±0.074 | 0.715±0.099 |
| CLIP + Random Drop | 0.741±0.073 | 0.735±0.097 |
| **HAF (GA + Random Drop)** | 0.745±0.065 | **0.739±0.092** |
| *Parameter-free aggregation baselines (on GA)* | | |
| GA + Mean Pool | 0.606±0.122 | 0.700±0.090 |
| GA + Sum Pool | 0.636±0.077 | 0.717±0.077 |
| GA + Max Pool | 0.649±0.101 | 0.702±0.106 |
| *Comparable multimodal methods* | | |
| PS3 | 0.626±0.123 | 0.718±0.117 |
| MDLM | 0.557±0.145 | 0.626±0.122 |
| MFMF | 0.675±0.089 | 0.732±0.127 |
| Bilinear Interaction | 0.682±0.101 | 0.684±0.119 |

### 4.3. Overall Quantitative Results

Table 1 organizes the results into pathology-only baselines, fusion models and HAF variants, parameter-free aggregation baselines, and representative multimodal methods. The official HANCOCK baselines (**WSI-CLAM**, **TMA-CLAM**, **WSI+TMA-CLAM**) provide a unimodal reference point, and our reproduction of **WSI+TMA-CLAM** closely matches the reported results (Accuracy 0.712, AUC 0.679). The **WSI+TMA Fusion**

**MLP** improves accuracy (0.739) but yields a lower AUC (0.668). Extending fusion to all seven modalities with the **All-Modality Fusion MLP** gives moderate improvements (AUC 0.694), although performance varies considerably across folds. Among the single-component variants, **Global Alignment (GA)** achieves the highest accuracy (0.752), while **CLIP Alignment** shows a similar but slightly weaker trend. **Random-Modality Drop** primarily improves AUC (0.715) relative to the All-Modality Fusion MLP. Combining Drop with alignment further increases discrimination: **CLIP + Drop** reaches an AUC of 0.735, and the full **HAF (GA + Drop)** achieves the best overall performance (AUC 0.739). To compare Random-Modality Drop with simpler parameter-free aggregation on aligned embeddings, we additionally evaluate mean/sum/max pooling on GA features. While these baselines are competitive in AUC (**GA+Sum Pool**: 0.717; **GA+Mean Pool**: 0.700; **GA+Max Pool**: 0.702), they are consistently below HAF (AUC 0.739) and yield lower accuracy (0.606–0.649). For comparison, representative multimodal frameworks such as **PS3** (0.626 / 0.718), **MDLM** (0.557 / 0.626), **MFMF** (0.675 / 0.732), and **Bilinear Interaction** (0.682 / 0.684) perform notably worse across accuracy and AUC, with HAF achieving consistent improvements in accuracy over all comparable methods and in AUC over all except MFMF.

### 4.4. Robustness to Modality Drop

Fig. 2 summarizes model robustness under varying drop probabilities during testing. AUC and accuracy remain nearly constant up to $\rho = 0.4$, indicating that the model compensates for missing modalities by relying on reliable inputs, and degrade only when most modalities are absent ($\rho > 0.4$), demonstrating improved resilience to real-world incompleteness and noise.

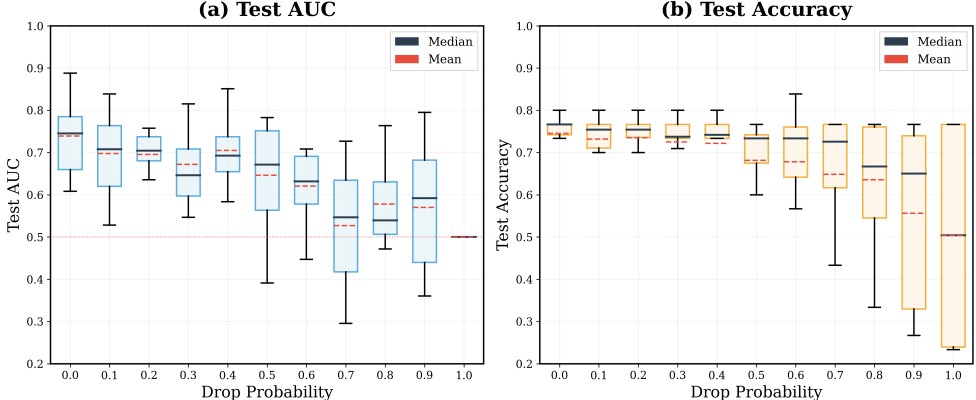

Figure 2: **Model robustness under modality dropout.**

### 4.5. Representation Analysis

To illustrate the alignment effect, Fig. 3 visualizes multimodal embeddings before and after global alignment. Each color represents one modality, and each point corresponds to a patient from the test set.

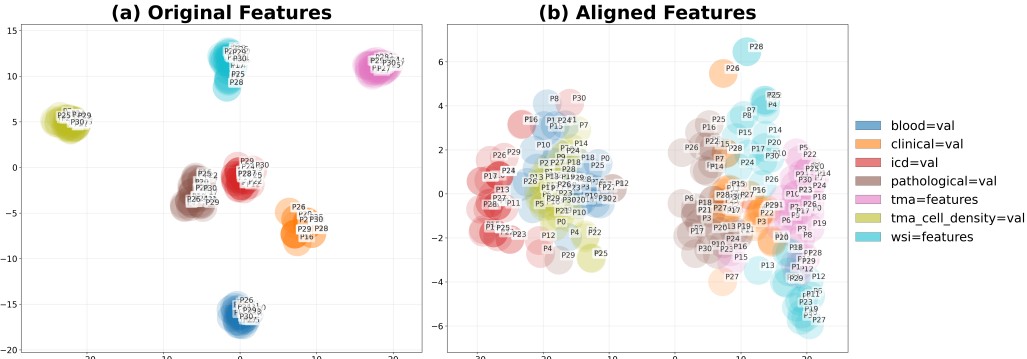

Figure 3: **t-SNE projection of multimodal embeddings before and after global alignment.** The horizontal and vertical axes represent the first and second t-SNE components, respectively.

Before alignment, modalities form well-separated clusters with large inter-modality gaps, while patient embeddings within the same modality are highly overlapping, indicating strong modality bias but limited patient discriminability. After alignment, modalities become more coherent along shared axes and different patients are pulled further apart, simultaneously reducing inter-modality discrepancies and enhancing inter-patient separability.

## 5. Discussion

This work demonstrates that reliable multimodal survival classification in clinical settings is limited when achieved by feature aggregation alone. Our results suggest that performance gains in HAF arise from explicitly addressing three structural challenges: (i) geometric incompatibility across modalities, (ii) systematic modality missingness, and (iii) cross-objective interference during end-to-end training. By decoupling representation learning, alignment, and fusion into staged objectives, HAF provides a framework for building robust multimodal classification.

While global alignment improves overall performance, the aligned representations do not collapse into perfect subject-level clusters. As shown in Fig. 3, a degree of modality-wise separation remains after alignment. This behavior is not contradictory to the objective of alignment. Rather than enforcing strict feature collapse across modalities, the alignment loss encourages shared low-rank structure while preserving modality-specific residual variations. In highly heterogeneous clinical data, such residual structure is in- evitable and may even be beneficial, as different modalities capture complementary aspects of patient state that cannot be fully reconciled in a single latent space.

The relatively weak performance of several literature baselines can be attributed to the heterogeneous quality of the available modalities (see Table 1). Methods that treat all modalities symmetrically, or that rely on weaker modalities as queries (e.g., attention-based fusion), may propagate noisy signals into the fusion process, while decision-level late fusion may discard cross-modal interactions before meaningful integration occurs. In contrast, global alignment yields more consistent results by placing heterogeneous modalities into

a comparable representation space, and multimodal integration provides systematic gains over all single-modality predictors (Table 2). HAF further improves over naive fusion, suggesting that combining alignment with robustness-aware fusion is beneficial in settings where modality quality varies substantially.

Random-Modality Drop explicitly trains the fusion module under missing-modality conditions, which aligns with the stable performance under moderate drop rates in Fig. 2. The detachment ablation suggests an optimization trade-off in multimodal alignment. Allowing downstream task gradients to directly update the alignment module can some- times improve raw discriminative metrics, but may introduce conflicts between modality- consistency objectives and task-specific objectives. Stage-wise detachment therefore serves as a regularization mechanism to decouple these objectives, leading to more stable shared representations (see Appendix C).

## 6. Limitations

Despite the empirical improvements observed across multiple settings, a limitation is that the global alignment objective does not guarantee that the resulting shared representation space is optimal for the downstream task. While alignment enforces cross-modality consistency, it does not explicitly optimize for task-specific separability, and in some cases may reduce inter-patient discriminability when the alignment geometry deviates from the task-relevant structure.

## 7. Conclusion

We presented **HAF**, a staged and detached early multimodal fusion framework that mitigates cross-objective interference, stabilizes pathology representations, establishes a shared cross-modality geometry, and improves robustness under missing or noisy modalities. Beyond its empirical gains on HANCOCK, HAF also provides a general recipe for designing principled fusion pipelines in highly heterogeneous clinical settings. Future work will focus on exploring stronger task-aware alignment, incorporating finer spatial reasoning, and validating HAF on broader clinical datasets. In addition, extending HAF to prospective cohorts and evaluating its utility in real clinical decision support will further illuminate its translational potential.

## 8. Acknowledgements

This work was partially supported by US National Science Foundation IIS-2412195, CCF-2400785, the Cancer Prevention and Research Institute of Texas (CPRIT) award (RP230363), the National Institutes of Health (NIH) R01 award (1R01AI190103-01) and Microsoft Accelerate Foundation Models Research (2024).

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

## Appendix A. Dataset Statistics and Evaluation Protocol

We conduct all experiments on the publicly available HANCOCK cohort (Dorrich et al., 2025). Patient-level survival labels are obtained from the official `target.csv` file released in the HANCOCK GitHub repository. The full cohort contains 763 patients with survival annotations. Among them, 701 patients have whole-slide images (WSI) available, corresponding to 1,078 WSI slides in total. We further restrict the cohort to patients with both WSI and TMA available, excluding 2 patients who have no TMA available, resulting in 699 patients used in our experiments. For the remaining five structured modalities, missing feature entries are imputed following the official HANCOCK preprocessing protocol released in the original repository. Modality-level missingness is simulated during training and evaluation via random-modality masking. Among these, 509 patients are labeled as living and 190 as deceased. We perform patient-level 10-fold cross-validation, where approximately 69–70 patients are held out for testing in each fold. All reported metrics are computed strictly at the patient level.

## Appendix B. Single Modality

Table 2: Single-modality baselines on the HANCOCK cohort. Each model uses CLAM for pathology-based modalities and a two-layer MLP for structured modalities. Results are reported as mean $\pm$ standard deviation over 10 patient-level folds.

| Model | accuracy (mean $\pm$ std) | AUC (mean $\pm$ std) |
|---|---|---|
| blood | $0.703 \pm 0.060$ | $0.644 \pm 0.111$ |
| clinical | $0.632 \pm 0.076$ | $0.603 \pm 0.117$ |
| icd | $0.626 \pm 0.106$ | $0.581 \pm 0.076$ |
| pathological | $0.665 \pm 0.091$ | $0.650 \pm 0.140$ |
| tma cell density | $0.645 \pm 0.106$ | $0.550 \pm 0.135$ |

## Appendix C. Ablations on Detachment

Training without detachment corresponds to end-to-end optimization across all stages, which often entangles heterogeneous objectives and distorts the geometry needed for modality substitutability. When heterogeneous objectives are trained jointly without isolation, cross-stage gradients can conflict and bias earlier representations toward downstream fusion losses. Detachment prevents such interference by allowing each stage to converge independently before passing non-trainable features forward. As shown in Table 3, comparing HAF with its non-detached variant demonstrates that detachment is beneficial for our framework, yielding a consistent improvement on the ranking-oriented objective (+1.8 pp AUC), while inducing only a marginal change in classification accuracy.

A small trade-off appears in the alignment-only setting: without detachment, global alignment is less complete and modality-specific biases leak into the shared space. These

Table 3: **Ablations on detachment.** "w" denotes training *with* detachment, "w/o" indicates no detachment.

| Setting | Metric | w (with detach) (mean ± std) | w/o (no detach) (mean ± std) |
|---|---|---|---|
| All modalities | accuracy | **0.748±0.046** | 0.738±0.063 |
| | AUC | 0.694±0.113 | **0.698±0.110** |
| + Global Alignment | accuracy | **0.752±0.047** | 0.752±0.050 |
| | AUC | 0.698±0.127 | **0.727±0.108** |
| + Random Drop | accuracy | **0.748±0.074** | 0.748±0.081 |
| | AUC | **0.715±0.099** | 0.714±0.101 |
| **HAF** | accuracy | 0.745±0.065 | **0.748±0.052** |
| | AUC | **0.739±0.092** | 0.721±0.098 |

residual biases can sometimes inflate AUC by exploiting cohort-specific correlations, yet they blur patient-level separability and marginally undermine generalization. In contrast, detachment removes such interference and yields more consistent, semantically grounded representations, improving the robustness of the full HAF framework.

## Appendix D.  Additional Representation Visualization

The heatmap in Fig. 4 provides an additional qualitative illustration of the alignment effect. Before alignment, feature intensity sequences across modalities are largely uncorrelated, whereas after alignment they become more synchronized for the same patient, indicating improved cross-modality consistency.

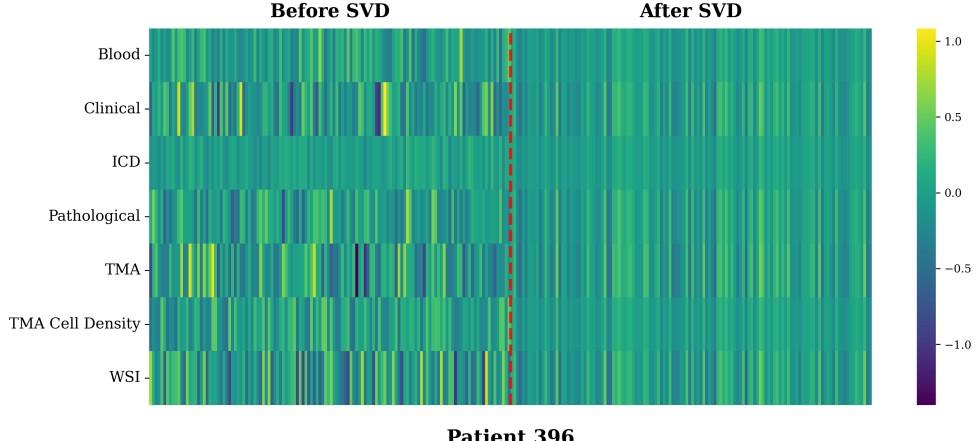

Figure 4: **Heatmap of aligned modality representations for a representative patient.**

## Appendix E. Statistical Significance

We conduct statistical significance testing using paired tests to compare our proposed model (HAF) against each baseline on the same patient-level test sets. For classification accuracy, we apply McNemar's test to paired predictions obtained from identical test folds, which is appropriate for discrete classification outcomes. For AUC, we apply DeLong's test for correlated ROC curves, as AUC reflects a ranking-based continuous statistic. We further apply the Holm–Bonferroni correction, which is a more powerful step-down procedure than the standard Bonferroni correction, to adjust the resulting p-values while controlling the family-wise error rate. The Holm–Bonferroni procedure is applied in a step-down manner by ordering comparisons according to ascending raw p-values and sequentially adjusting rejection thresholds. All tests use a significance level of $\alpha = 0.05$. The set of comparable methods (baselines) includes: PS3, MDLM, MFMF, and Bilinear Interaction.

Table 4: Statistical significance testing (HAF vs. baselines) for classification accuracy

| Baseline | Raw $p$-value | Holm-adjusted $p$ | Significant ($\alpha = 0.05$) |
|---|---|---|---|
| MDLM | 0.00078 | 0.00312 | Yes |
| MFMF | 0.00860 | 0.02580 | Yes |
| PS3 | 0.01310 | 0.02620 | Yes |
| Bilinear Interaction | 0.01870 | 0.02620 | Yes |

Table 5: Statistical significance testing (HAF vs. baselines) for AUC

| Baseline | Raw $p$-value | Holm-adjusted $p$ | Significant ($\alpha = 0.05$) |
|---|---|---|---|
| MDLM | 0.00063 | 0.00252 | Yes |
| Bilinear Interaction | 0.00930 | 0.02790 | Yes |
| PS3 | 0.02340 | 0.04680 | Yes |
| MFMF | 0.09549 | 0.09549 | No |

