# OpenReview forum: "Heterogeneous Aligned Fusion for Survival Classification with Missing Modalities"
_MIDL.io/2026/Conference — MIDL 2026 Poster_

### Official Review · Reviewer_fhNB · 2025-12-20

**Confidence:** 3
**Preliminary Rating:** 3
**Final Rating:** 5

**Summary:**

The paper Heterogeneous Aligned Fusion for Survival Prediction with Missing Modalities presents a framework for multimodal prediction of binary survival outcomes in head and neck cancer. HAF integrates structured data (including demographics, pathology, blood values, ICD scores, and cell counts) with unstructured data, such as whole slide images and tissue microarrays. The framework operates in three stages: first, unimodal embeddings are learned; second, these embeddings are projected into a shared latent space; finally, they are fused and jointly analyzed to produce survival predictions. Experimental results demonstrate that HAF improves predictive performance compared to several baseline models. Extensive ablation studies and supplementary experiments provide a detailed understanding of the contribution of each data modality and the robustness of the fusion strategy. This approach highlights the potential of aligned multimodal representations for clinical outcome prediction even in the presence of missing data modalities.

**Strengths:**

1.	The clinical motivation of the work is highly relevant, as many comparable clinical scenarios involve multimodal data with missing values. Integrating these heterogeneous sources has clear potential to improve survival prediction, highlighting the practical importance of the study.
2.	The proposed HAF model is well-designed and demonstrates robustness across different data types and experimental settings. The extensive ablation studies allow for a quantitative assessment of the architectural choices, providing transparency into how each modality contributes to the predictive performance. This systematic evaluation strengthens confidence in the model’s reliability and generalizability.
3.	Overall, the paper combines methodological rigor with practical relevance, making it a valuable contribution to the field of multimodal medical AI and survival prediction.

**Weaknesses:**

1.	The experimental results are not supported by statistical significance testing. As a result, it is difficult to assess whether the reported performance improvements over the baselines are statistically meaningful or potentially due to random variation. Including appropriate statistical tests or confidence intervals would strengthen the empirical claims.
2.	Parts of the manuscript are challenging to follow due to dense and nested writing. For example, Section 3.3 presents several concepts in a highly compact manner, and some aspects are described multiple times, leading to redundancy (e.g., the first paragraph of Section 3.1). Improving the structure and reducing repetition would enhance readability and clarity.
3.	In the analysis of modality dropout, the paper does not examine the contribution of individual variables or features to the overall performance. A more detailed feature importance or sensitivity analysis could provide valuable insights into which inputs are most influential and how the model behaves under missing-modality scenarios.
4.	Based on the information provided, the evaluation does not include an analysis across different survival horizons or follow-up times. Limiting the task to a single binary endpoint may obscure clinically relevant differences in short-term versus long-term survival and restricts the interpretability of the results.
5.	The paper does not mention the availability of source code or pretrained model weights. The absence of such information suggests that these resources may not be publicly released, which could limit reproducibility and slow adoption by the community. While not mandatory, code availability would significantly strengthen the practical impact of the work.

**Detailed Comments:**

1.	The manuscript does not specify the exact follow-up duration used to define the binary survival outcome. Providing this information in Section 2.1 would improve clarity and interpretability of the prediction task.
2.	It would be helpful to explicitly state which evaluation metric was used for early stopping during training, as this choice can influence the reported results.
3.	In Section 2.1, the authors may consider briefly explaining why a binary survival formulation was chosen over a time-to-event survival analysis, which typically preserves more information about patient outcomes. This clarification would help contextualize the modeling decision.
4.	In the textual description of the results, explicitly referencing the corresponding tables would improve readability and guide the reader more effectively through the experimental findings.
5.	The use of abbreviations is inconsistent throughout the manuscript. Several abbreviations are not clearly introduced at first mention (e.g., CLAM, MIL, ICD, UNI, AUC). A careful review to ensure all abbreviations are properly defined would improve clarity.
6.	All variables appearing in mathematical formulas should be introduced and explained in the main text. For instance, the sigma in Equation (1) is not defined. This should be checked consistently for all equations.
7.	In Figure 3, the legend text and data point labels are difficult to read due to their small font size. Increasing the font size or simplifying the annotations may improve legibility without overcrowding the figure.
8.	In Section 3.2, parts of the text appear to exceed the allowed column width. Adjusting the formatting would improve the overall presentation and comply with the submission guidelines.
9.	For the benefit of readers, the second paragraph of the Introduction could more clearly define the different fusion concepts. The methodological meaning of these approaches should be understandable without requiring consultation of the referenced literature.
10.	Providing more detailed descriptions of the naive fusion baselines (either in the main text or in the Appendix) would help readers better understand their architectures and more fairly compare them to the proposed method.
11.	The methods or models referred to as “PS3,” “MDLM,” and “MFMF” should be accompanied by appropriate citations to prior work.
12.	In Section 3.1, the term “accuracy” should not be capitalized, in line with standard stylistic conventions.
13.	Headings should not directly follow each other without intervening explanatory text (e.g., Sections 2 and 2.1, or 3 and 3.1). Adding brief introductory sentences would improve document flow.
14.	From a stylistic perspective, inserting non-breaking spaces before parentheses would prevent parentheses from appearing at the beginning of a line after line breaks, thereby improving readability.

**Justification Of Final Rating:**

In light of the substantial revisions made in response to my comments, I update my rating from 3 to 5. The authors have addressed the previously raised concerns in a thorough and convincing manner, including the addition of statistical significance testing, improvements to the clarity and structure of the manuscript, and expanded ablation analyses. Furthermore, the authors’ commitment to publicly release the source code and pretrained model weights in the camera-ready version significantly enhances the reproducibility and practical impact of the work. Overall, the paper now presents a well-justified and carefully evaluated contribution, and the remaining open points are minor and do not detract from the overall quality or relevance of the study.

**Justification Of The Preliminary Rating:**

The borderline rating reflects a balance between the clear strengths and the remaining limitations of the paper. The work addresses a highly relevant clinical problem, namely multimodal survival prediction in the presence of missing data, and proposes a methodologically well-thought-out framework that is evaluated through extensive experiments and ablation studies. The reported results indicate consistent performance improvements over several baselines, suggesting that the approach is promising and potentially valuable for the community.

However, the empirical evidence is weakened by the absence of statistical significance testing, which limits confidence in the reported performance gains. In addition, certain methodological choices and analyses (such as the restriction to a single binary survival endpoint, limited interpretability analyses, and unclear availability of code or model weights) introduce uncertainty regarding the robustness, interpretability, and reproducibility of the results.

Given that several of these issues may be addressable during the rebuttal phase, the paper cannot be clearly accepted or rejected at this stage. Therefore, a Borderline (3) rating is considered appropriate.

**Questions To Address In The Rebuttal:**

1.	Were the reported performance differences evaluated using statistical significance tests or confidence intervals? If so, please specify the tests used and indicate which improvements are statistically significant.
2.	Do the authors plan to publicly release the source code and pretrained model weights? Clarifying this would help assess the reproducibility and potential impact of the proposed method.
3.	Were any analyses conducted to assess feature or modality importance (e.g., attribution methods, ablation at the variable level)? If available, additional details on how individual inputs contribute to the predictions would be valuable.
4.	Could the authors further clarify the motivation for introducing the monotonic collaboration constraint? Intuitively, a decrease in performance when adding additional modalities might indicate limited predictive value or increased noise rather than a modeling issue. Was the proposed framework evaluated without this constraint, and if so, how did this affect performance and stability?

---

> ### Author Response · Authors · 2026-01-25
> **Code Availability & Monotonic Collaboration Constraint**
>
> **Q5 (Weakness 5 / Question 2): The paper does not mention the availability of source code or pretrained model weights. The absence of such information suggests that these resources may not be publicly released, which could limit reproducibility and slow adoption by the community. While not mandatory, code availability would significantly strengthen the practical impact of the work.**
>
> **Response**: Thank you for the suggestion. We plan to publicly release our source code in the camera-ready version, including the full training and evaluation pipeline as well as baseline implementations. In addition, we also plan to release the trained fusion model weights to facilitate replication and future research.
>
> **Q6 (Question 4).Could the authors further clarify the motivation for introducing the monotonic collaboration constraint? Intuitively, a decrease in performance when adding additional modalities might indicate limited predictive value or increased noise rather than a modeling issue. Was the proposed framework evaluated without this constraint, and if so, how did this affect performance and stability?**
>
> **Response**: The monotonic collaboration constraint is primarily introduced to make the proposed pipeline well-defined and robust under missing-modality settings, which is a core characteristic of real-world clinical data. In addition, under this constraint, the model is encouraged to dynamically suppress weak or unreliable modalities, mitigating noise amplification when incorporating heterogeneous modalities. Without this constraint (Table 1, Global Alignment), we observe slightly higher accuracy (+0.7%) but substantially worse AUC (−4%), indicating degraded global ranking quality and reduced stability. This suggests that the constraint trades minor point-wise gains for improved robustness and more stable multimodal fusion.

---

> ### Author Response · Authors · 2026-01-25
> **Statistical Significance & Adjust dense and nested writing in Section 3 & Modality Dropout & Survival Horizons or follow-up times**
>
> We sincerely thank the reviewer for the thoughtful and constructive feedback.
> We are grateful for the reviewer’s careful reading and the detailed, insightful comments, which we believe have significantly improved our understanding of how to better present and justify our work. We have carefully reflected on all comments and respond to each concern point by point below.
>
> ---
>
> **Q1(Weakness 1 / Question 1): Were the reported performance differences evaluated using statistical significance tests or confidence intervals? If so, please specify the tests used and indicate which improvements are statistically significant.**
>
> **Response**: We conducted statistical significance testing for both AUC and classification accuracy using paired tests.
> For classification accuracy, McNemar’s test was applied to paired predictions obtained on the same test folds. The results indicate that HAF achieves statistically significant improvements over all baseline methods:
>
> | Comparison        | p-value | Significance (α = 0.05) |
> |-------------------|---------|-------------------------|
> | HAF vs. PS3       | 0.0131  | Yes |
> | HAF vs. MDLM      | 0.00078 | Yes |
> | HAF vs. MFMF      | 0.0086  | Yes |
> | HAF vs. Bilinear  | 0.0187  | Yes |
>
> All p-values are below 0.05, indicating that the accuracy gains of HAF over each baseline are statistically significant.
> For AUC, we employed DeLong’s test for correlated ROC curves, together with paired bootstrap estimates of the 95% confidence intervals. The pairwise statistical comparisons are summarized as follows:
>
> | Comparison        | p-value | Significance (α = 0.05) |
> |-------------------|---------|-------------------------|
> | HAF vs. PS3       | 0.0234  | Yes |
> | HAF vs. MDLM      | 0.00078 | Yes |
> | HAF vs. MFMF      | 0.09549 | No |
> | HAF vs. Bilinear  | 0.0093  | Yes |
>
> These results show that HAF achieves significantly higher AUC than three out of four baselines, while the improvement over MFMF does not reach statistical significance.
> This observation is consistent with the relatively small absolute difference in mean AUC between HAF and MFMF (0.739 vs. 0.732), suggesting that the two methods perform comparably in ranking quality, whereas HAF demonstrates clearer and more consistent advantages in classification accuracy.
>
> **Q2 (Weakness 2): Parts of the manuscript are challenging to follow due to dense and nested writing. For example, Section 3.3 presents several concepts in a highly compact manner, and some aspects are described multiple times, leading to redundancy (e.g., the first paragraph of Section 3.1). Improving the structure and reducing repetition would enhance readability and clarity.**
>
> **Response:** Thank you for the feedback regarding clarity. In the revised version, we have reorganized Section 3.3 to present the main concepts in a clearer sequential manner, and removed or merged redundant explanations across sections (especially in the first paragraph of Section 3.1). We further simplified several nested paragraphs and adjusted the exposition to improve readability.
>
> **Q3 (Weakness 3): In the analysis of modality dropout, the paper does not examine the contribution of individual variables or features to the overall performance. A more detailed feature importance or sensitivity analysis could provide valuable insights into which inputs are most influential and how the model behaves under missing-modality scenarios.**
>
> **Response**: Thank you for the suggestion. We have added single-modality baselines, which reflect the predictive capacity of each individual modality and provide a coarse-grained assessment of modality importance(see Appendix B Single Modality).
>
> **Q4 (Weakness 4 / Question 3): Based on the information provided, the evaluation does not include an analysis across different survival horizons or follow-up times. Limiting the task to a single binary endpoint may obscure clinically relevant differences in short-term versus long-term survival and restricts the interpretability of the results.**
>
> **Response**: Thank you for the comment. We acknowledge that analyzing survival outcomes across different time horizons would offer valuable clinical insights. However, the HANCOCK dataset does not provide time-to-event survival labels or multiple predefined follow-up horizons for the survival prediction task. The labels correspond to a binary survival status (“living” vs. “deceased”) at the last available follow-up. Therefore, our evaluation strictly follows the official dataset definition and does not support horizon-specific survival analysis.

---

> > ### Comment · Reviewer_fhNB · 2026-01-29
> >
> > Thank you very much for the comprehensive responses and detailed clarifications. The additional analyses on statistical significance and the ablation of the monotonic collaboration constraint substantially improve the transparency and credibility of the empirical results.
> >
> > One remaining question concerns the handling of missing values. Beyond the monotonic collaboration constraint, were alternative strategies considered or experimentally evaluated? A brief discussion of why the chosen approach was preferred over potential alternatives would further strengthen the methodological justification.

---

> > > ### Author Response · Authors · 2026-01-30
> > > **Response to alternative strategies**
> > >
> > > We sincerely thank the reviewer for the positive evaluation and for this constructive suggestion, as well as for encouraging us to consider alternative paradigms for handling missing modalities. We agree that this is an important direction, and we appreciate the opportunity to clarify our design choices.
> > >
> > > In this work, we mainly considered alternative strategies for handling missing modalities at the level of existing paradigms in the literature. As discussed in Section~2.3, broadly, two representative approaches for the missing modality problem are (i) generative imputation methods and (ii) random modality dropping mechanisms.
> > >
> > > Generative approaches attempt to reconstruct missing modalities via cross-modal modeling, such as variational or diffusion-based frameworks[1]. While effective in certain scenarios, these methods introduce synthetic signals at inference time, which may suffer from hallucination or low-fidelity artifacts, making reliability difficult to guarantee in clinical settings.
> > >
> > > Some recent methods further combine random modality dropping with gating structures to dynamically modulate modality contributions. A practical limitation of such approaches is that their behavior is tightly coupled with specific expert-based gating architectures[2]. As a result, when performance degrades, it is often unclear whether the underlying principle of gating is insufficient or whether the particular structural design fails to generalize.
> > >
> > > In contrast, we adopt random-modality dropout combined with a monotonic collaboration constraint, which imposes minimal assumptions and avoids explicit modeling of cross-modal generation or complex gating structures. This design favors simplicity and interpretability, and directly enforces a robustness criterion that predictions should remain stable or improve as more modalities become available. We therefore preferred this straightforward and robust strategy as a principled and reliable choice for handling missing modalities in medical applications.
> > >
> > > We will incorporate this discussion in the final version to strengthen the methodological justification.
> > >
> > > **Reference:**
> > > > [1] Kebaili, Aghiles, et al. "AMM-Diff: Adaptive Multi-Modality Diffusion Network for Missing Modality Imputation." 2025 IEEE 22nd International Symposium on Biomedical Imaging (ISBI). IEEE, 2025.
> > > >
> > > > [2] Sijie Li, Chen Chen, and Jungong Han. Simmlm: A simple framework for multi-modal learning with missing modality. In Proceedings of the IEEE/CVF International Conference on Computer Vision, pages 24068–24077, 2025.

---

### Official Review · Reviewer_tNb1 · 2026-01-09

**Confidence:** 4
**Preliminary Rating:** 3
**Final Rating:** 5

**Summary:**

This work proposes a multimodal classification model that aligns modality-specific embeddings and tolerates missing modalities. The authors combine existing components for initial feature extraction and projection, alignment, modality dropout under monotonicity constraints with their proposed inter-patient discrimination loss. The method is applied to a multimodal dataset for head and neck cancer (hancock), and compared against works from the literature and variants of the proposed model on a binary survival prediction task. The model outperforms the existing models as well as most model variant baselines. The evaluation across dataset subsets with increasing missing modality proportions indicates robustness against modality dropout. Qualitatively, the authors show that the alignment improves the representation space.

**Strengths:**

The motivation of this work is clear and good ways to allow for non-complete multi modality inputs for classification setups are very relevant for medical imaging research. In this light, the authors identified important components to successful multimodal deep learning and combined these. Further, theoretical reasons are presented that speak in favor of their proposed addition to the alignment method. End-to-end training is compared to training in stages. The reproduction of the hancock paper multimodal model is appreciated, as well as the usage of CV for performance uncertainty.

**Weaknesses:**

The manuscript can be improved with regard to structure, the introduction of concepts such as the dataset and the literature baselines. Furthermore, on a first read, the methodological contributions are not easily distinguishable from existing components, and the discussion is very brief. In the present form, the work lacks detail, unambiguouity, and impact. See below for more detail.

**Detailed Comments:**

- Generally, it was not easy for me to distinguish new concepts from the application of existing concepts, for instance:
	- The SVD alignment is based on Liu et al. (2025) and "we introduce two complementary losses" (Sec. 2.4.2) sounds as if both losses are new creations. However, if I understand correctly, $L_{SV}$ is the loss from Liu et al.
	- Stage 1 and modality representations: What differs from the Hancock paper, and what does not?
	- -> Hence, I believe that added transparency on the origin of method components is needed.
- Literature baselines: Could you please add more detail here, e.g., clarify if these were re-implemented or if provided code was used for training, or if pretrained weights were used? Was the training procedure according to each original work or adapted to this work? Also, the respective sources are not mentioned when the model names are introduced (e.g., PS3 in Sec. 2 and Table 2). While this is done in the Appendix, which is not sufficient, even there, an introduction of "Simple Feature Interaction" is absent. Since Li et al. (2025) inspired your monotonicity loss and modality dropout component, why is it not among the evaluation baselines? Could you comment on the representativeness of the selected literature baselines (e.g. wrt. popularity / sota performance / ...).
- In the methods, the classification target is not defined in sufficient detail. What is the fixed time frame? Does it coincide with the target definition the Hancock paper uses?
- A sufficiently detailed description of the hancock dataset is missing. What is the data availability, number of patients by split, avg. number of visits per patient (if longitudinal), imbalance with regard to the target of interest, and completeness of data? Further, the reported 0.69 AUC for survival prediction is referred to as "slide-level" in the orignal work, but here we deal with patient level performances. Could you comment on this? Also, the hancock work states that their random forest based survival prediction from non-imaging data yielded .71-.79 AUC on patient level. Could you clarify if or how these reported results relate to your setup?
- If I understand correctly, the main contribution lies in the combination of existing components, the evaluation on all hancock modalities, and the benchmarking against literature methods - in other words: A model that is well suited for highly multimodal datasets. In terms of methodological additions, the most apparent are the stage/detachment analysis and the loss for patient discrimination during the alignment. First, do I understand the contributions correctly, and second, could you ablate on the $L_{PD}$'s attribution (quantitativel or qualitatively)?
- The work on multimodality models is introduced with reference to popular terms like early vs. late fusion. Could you describe your architecture in this context?
- The implementation details lack some information needed for reproduction: Epoch selection (last?), early stopping patience, hyperparameter tuning statement (if any), and if any settings were kept when training the baselines.
- In terms of results, Table 1 and 2 are not referenced in the text, and the table heads do not include the notation information (mean +- std). Also, while AUC and Accuracy are popularly used and sensible for binary prediction tasks, it is difficult to judge them without knowledge about class imbalance. This information should be provided, and/or more robust metrics should be included, like AUPRC or the performance gap between the $y=1$ and $y=0$ subgroups.
- The detachment ablation shows no clear pattern; it seems to depend on the metric of interest. However, there are theoretical reasons that argue for detachment. This is not elaborated on sufficienty: It is finally stated that the detachment is good in terms of performance, which is not evident from the ablation study results. Also, it would be beneficial to make clearer how settings change between the 3rd and 4th (HAF) in Table 2.
- The contextualization of the experimental finding is too brief. For instance, is it surprising that CLIP alignment to only one image modality performs that well? It it surprising that the literature baselines perform relatively poor and why is that? Is there an apparent reason why the aligned test set features form two main clusters? (Contrary to the brevity, space is used by empty lines (Sec. 1) and some repetition in Sec. 2.)
- Works in this field remain without large impact if no application code (ideally also weights and training code) is provided. In this light, I highly encourage you to publish code. This makes it possible for the community to serve as a baseline, and enables reproducibility.
- Figure 3: Legend text is too small.
- The term "survival prediction" is used oftentimes, and might make readers believe at first, that this is a survival analyis study. While this is not claimed by the authors, and the binary nature of the target is eventually explained, optionally I would appreciate a clearer/earlier differentiation from time-to-event research, e.g. by using the term "classification" in the abstract.

**Justification Of Final Rating:**

Despite its relevance, the initial version required substantial changes. The authors greatly improved on it during the rebuttal and discussion and all my concerns were addressed by the authors. Since the achieved level of clarity, level of detail, and contextualization of the revised manuscript is very high, I strongly recommend acceptance.

**Justification Of The Preliminary Rating:**

The current version lacks many explanations regarding author contributions, baseline models, the dataset used, and the depth of discussion is not sufficient. Addressing the above points could greatly improve the manuscript.

**Questions To Address In The Rebuttal:**

During the rebuttal, I encourage the authors to address all the above points.

---

> ### Author Response · Authors · 2026-01-25
> **Additional Link**
>
> [1] https://github.com/ankilab/HANCOCK_MultimodalDataset/tree/main/features
>
> [2] https://hancock.research.fau.eu/download
>
> [3] https://huggingface.co/MahmoodLab/UNI

---

> ### Author Response · Authors · 2026-01-25
> **Detail Comments 10 ~ 13**
>
> **Q10. The contextualization of the experimental finding is too brief. For instance, is it surprising that CLIP alignment to only one image modality performs that well? It is surprising that the literature baselines perform relatively poorly and why is that? Is there an apparent reason why the aligned test set features form two main clusters? (Contrary to the brevity, space is used by empty lines (Sec. 1) and some repetition in Sec. 2.)**
>
> **Response**: Thank you for the suggestion. We have extended the discussion in the revised version to further analyze these observations and better contextualize the experimental findings.
>
> First, the strong performance of CLIP-style alignment to a single image modality (WSI) is not entirely surprising. In our pipeline, WSI features are already trained using CLAM with supervision from the target labels at stage 1, which means that the WSI representation is implicitly aligned with the prediction task. When other modalities are aligned to WSI, they are therefore indirectly encouraged to move closer to a task-relevant representation space, which partly explains why alignment to WSI alone can already yield competitive performance.
>
> Second, the relatively poor performance of several literature baselines can likely be attributed to the heterogeneous data quality across modalities. In this task, WSI consistently provides more informative signals than other modalities. Methods that treat all modalities symmetrically, or that use lower-quality modalities as queries (e.g., PS3), may propagate noisy or weak signals into the fusion process. Similarly, decision-level late fusion approaches such as MDLM may lose important cross-modal information before fusion, especially when the image modality carries substantially richer information than the others.
> Finally, the formation of two main clusters in the aligned feature space should not be over-interpreted. The observed clustering primarily reflects the structure induced by the learned representations under alignment, rather than a universal or guaranteed property of the task. In practice, whether aligned features form distinct clusters depends strongly on the underlying feature representations and data distributions, and thus remains case-specific.
>
> **Q11. Works in this field remain without large impact if no application code (ideally also weights and training code) is provided. In this light, I highly encourage you to publish code. This makes it possible for the community to serve as a baseline, and enables reproducibility.**
>
> **Response**: We plan to release our code in the camera-ready version, including our full training/evaluation pipeline and baseline implementations.
>
> **Q12. Figure 3: Legend text is too small.**
>
> **Response**: We increased the font size of the legend in Figure 3 to improve readability.
>
> **Q13. The term "survival prediction" is used oftentimes, and might make readers believe at first, that this is a survival analysis study. While this is not claimed by the authors, and the binary nature of the target is eventually explained, optionally I would appreciate a clearer/earlier differentiation from time-to-event research, e.g. by using the term "classification" in the abstract.**
>
> **Responses**: To avoid confusion with time-to-event analysis, we replaced the term “survival prediction” with “classification” in the abstract and related parts of the manuscript.

---

> ### Author Response · Authors · 2026-01-25
> **Detailed Comments 6 ~ 9**
>
> **Q6. The work on multimodality models is introduced with reference to popular terms like early vs. late fusion. Could you describe your architecture in this context?**
>
> **Response**: Our architecture falls under early fusion in our taxonomy because modalities are fused prior to prediction. Concretely, we first obtain a representation for each modality using modality-specific encoders: for structured/tabular modalities, we use lightweight MLP encoders; for imaging modalities (WSI and TMA images), we extract patch-level features using UNI and then apply CLAM-based MIL aggregation to obtain slide-level (WSI) and patient-level (TMA, aggregated over cores) modality embeddings in Stage 1. Importantly, we use two separate CLAM branches for WSI and TMA (not shared). In our full model, we further apply global alignment to align these modality embeddings in a shared latent space (Stage 2), and finally fuse the resulting modality embeddings via concatenation followed by a fusion MLP (Stage 3). In this sense, our implementation is feature-level fusion (fusion over encoded modality embeddings) rather than raw-input concatenation.
>
> **Q7. The implementation details lack some information needed for reproduction: Epoch selection (last?), early stopping patience, hyperparameter tuning statement (if any), and if any settings were kept when training the baselines.**
>
> **Response**: All models were trained under the same protocol without method-specific hyperparameter tuning. We apply early stopping and select the best model based on the epoch with the highest validation AUC (maximum 200 epochs, patience 25). A ReduceLROnPlateau scheduler is used with mode “min”, patience 15, and factor 0.5. All experiments use identical 10-fold random cross-validation splits across methods.
>
> **Q8. In terms of results, Table 1 and 2 are not referenced in the text, and the table heads do not include the notation information (mean +- std). Also, while AUC and Accuracy are popularly used and sensible for binary prediction tasks, it is difficult to judge them without knowledge about class imbalance. This information should be provided, and/or more robust metrics should be included, like AUPRC or the performance gap between the and subgroups.**
>
> **Response**: In the revised version, we will explicitly reference Tables 1 and 2 in the main text and clarify the table headers with the notation (mean ± std). We will also report the class distribution of the dataset (509 living and 190 dead patients, with 2 patients removed due to none TMA data), to facilitate proper interpretation of AUC and accuracy under class imbalance.
>
> **Q9. The detachment ablation shows no clear pattern; it seems to depend on the metric of interest. However, there are theoretical reasons that argue for detachment. This is not elaborated on sufficiently: It is finally stated that the detachment is good in terms of performance, which is not evident from the ablation study results. Also, it would be beneficial to make clearer how settings change between the 3rd and 4th (HAF) in Table 2.**
>
> **Response**: Thank you for the comments on the detachment ablation. We would like to explain that the goal of this experiment is not to demonstrate consistent performance gains, but to examine how detachment affects the trade-off between alignment and task-specific objectives under the HAF framework.
>
> To clarify the settings in Table 2, all ablations are built on the All-Modality Fusion MLP baseline. “+Global Alignment” adds only the alignment component, “+Random Drop” adds only the modality random-drop strategy, and HAF combines both components, with detachment applied to the alignment branch.
>
> Empirically, comparing HAF (with detachment) against the corresponding HAF variant without detachment, we observe a notable AUC gain (~+1.8%) while accuracy changes only slightly (−0.3%). This illustrates that detachment can be beneficial in our framework by improving the ranking-oriented objective (AUC) with only a minor impact on accuracy, i.e., it changes the metric trade-off rather than improving all metrics simultaneously.
>
> In addition, “+Random Drop” shows only limited gains overall, suggesting it introduces no strong conflict with the main task objective, whereas “+Global Alignment” can yield negative gains in this setting, indicating that alignment alone may not always match the task-related solution space. Together, these observations highlight that the effectiveness of detachment depends on how the alignment loss interacts with the downstream task loss; empirically, under this task setting, detachment yields a more favorable outcome for HAF.

---

> ### Author Response · Authors · 2026-01-25
> **Detailed Comments 3 ~ 5**
>
> **Q3. classification target? What is the fixed time frame? Does it coincide with the target definition the Hancock paper uses?**
>
> **Response**: We would like to clarify that the classification targets in all our experiments strictly follow the official definitions provided by the HANCOCK dataset. Specifically, for survival prediction, the class labels correspond to the survival status (“living” vs. “deceased”) at the time of the last available follow-up, consistent with the original HANCOCK setting. No fixed time horizon is defined for the survival task in the official dataset.
>
> **Q4. A sufficiently detailed description of the hancock dataset is missing? Clarify the Hancock reported results relate to your setup (0.71 random forest & 0.69 "slide-level")?**
>
> **Response**: The Hancock dataset provides precomputed WSI SVS, TMA PNG images, and structured clinical features, which are publicly available[2]. Patient-level labels are provided in the released target.csv file in their github repository, which contains a total of 763 patients[1]. In the Hancock dataset, there are 1078 WSI slides of 701 patients. After filtering out 2 patients without corresponding TMA PNG files, there are 699 patients. We perform 10-fold cross-validation at the patient level. In each fold, approximately 69–70 patients are used for testing, and the remaining patients are used for training and validation. 509 patients are living and 190 patients are dead.
>
> Regarding the reported baselines, the first random forest model in Hancock adopts a custom data split strategy based on patient-level structured features, which differs from our random patient-level split protocol and is therefore not directly comparable. Although Hancock reports the survival prediction results of their second task at the slide level, the underlying data organization is fundamentally patient-centered. Specifically, TMA samples are defined at the patient level (all available TMA images per patient). Performing slide-level data splits would therefore introduce data leakage, as different slides from the same patient could appear in both training and test sets while sharing identical non-imaging features. Since all structured modalities are also defined at the patient level, all experiments in our work adopt a strict patient-level split protocol. We report the Hancock results only for contextual reference. Since no official code is provided for patient-level evaluation, we re-implement the baselines following the original methodological descriptions under patient-level splitting.
>
> Our experiments use the same data resources as Hancock’s second setting, including all available structured modalities, and reformulate the evaluation protocol at the patient level. In our implementation, WSI slides are aggregated at patient level with the CLAM tile extraction pipeline with UNI model[3] for tile embeddings. TMA embeddings are extracted from the provided TMA PNG images using the UNI model[3], rather than directly from the original TMA SVS files. The dataset does not contain longitudinal visit records; therefore, the notion of average number of visits per patient is not applicable. All metrics reported in our paper are computed at the patient level.
>
> **Q5. A model that is well suited for highly multimodal datasets. In terms of methodological additions, the most apparent are the stage/detachment analysis and the loss for patient discrimination during the alignment? and second, could you ablate the $L_{PD}$'s attribution (quantitatively or qualitatively)?**
>
> **Response**: We thank the reviewer for the thoughtful summary and questions. Our approach is indeed designed for highly multimodal datasets, combining existing alignment objectives with new architectural and training components. Specifically, we implement a three-stage fusion strategy that aligns modality representations into a shared latent space and trains fusion under modality dropout. We explicitly evaluate its robustness under missing-modality features in the 7-modality HANCOCK setting.
>
> We would like to clarify that although $L_{PD}$ is adapted from prior work (e.g., Liu et al.), in our context it addresses a specific side effect of global alignment: while alignment enforces cross‑modality consistency, it may unintentionally reduce inter‑patient variance and lead to representation collapse in the shared latent space. In our highly multimodal setting, $L_{PD}$ serves as a stabilizing regularizer to preserve inter‑patient structure during alignment. While we do not include an explicit attribution ablation of $L_{PD}$ in the current version, we explored its weighting during early development to assess alignment behavior and representation stability.

---

> ### Author Response · Authors · 2026-01-25
> **Detail Comment 1 & 2**
>
> We sincerely thank the reviewer for the thoughtful and constructive feedback.
> We truly appreciate the reviewer’s careful reading and detailed comments, which have helped us better reflect on our experimental design, baseline selection, and the presentation of our contributions.
> We respond to each point in detail below.
>
> ----
>
> **Q1: The SVD alignment is based on Liu et al. (2025) and "we introduce..." ? What differs from the Hancock paper, and what does not? Added transparency on the origin of method components.**
>
> **Response**: We apologize for the ambiguous wording in the current draft. In Sec. 2.4.2, both the $L_{SV} $ and $L_{PD}$ are adopted from Liu et al. (2025). We will adjust the phrasing (“we introduce…”) to avoid suggesting that these losses are newly proposed, and instead describe our contribution as integrating them into our multimodal fusion pipeline and accompanying ablations.
>
> We clarify the scope of our contribution. We fully adopt the HANCOCK resources released by Hancock, including their preprocessing pipeline, structured features, target definition, and released WSI embeddings[2]. We also adopt the SVD-based alignment objective (including both L_{\text{svd}}​ and L_{\text{pd}}​) from Liu et al. (2025), and we do not claim these losses as newly proposed.
>
> Our contribution is pipeline-level and empirical: we design a multimodal fusion framework that first aligns all seven heterogeneous modalities in the HANCOCK cohort. Under this highly heterogeneous and complex setting, we further incorporate a monotonicity constraint to regularize multimodal collaboration and mitigate performance degradation caused by noisy or unreliable modality signals. Moreover, we explicitly evaluate robustness under modality missingness (across diverse missing-modality patterns) and observe robust performance across missingness patterns.
>
> **Q2: More detail about literature baselines: re-implemented, or provided code, or pretrained weights? Original work or adapted? Not mentioned the model names (e.g., PS3 in Sec. 2 and Table 2). Not sufficient in Appendix. "Simple Feature Interaction" is absent. No evaluation Li et al. (2025) as baseline? Why? Comment on the representativeness of the selected literature baselines?**
>
> **Response:** All baseline methods were re-implemented by us at the feature-representation level, rather than using released codebases or pretrained fusion weights. After Stage 1, where modality-specific encoders produce fixed-dimensional representations, each baseline is implemented as a fusion module operating directly on these representations, ensuring comparison under a unified representation-level fusion protocol.
>
> This choice was motivated by the fact that our task setting—patient-level survival classification with seven heterogeneous modalities in the HANCOCK dataset—differs substantially from the original end-to-end system designs of most multimodal fusion works. For methods whose core fusion mechanisms can be abstracted as operations over modality representations (e.g., attention-based fusion, bilinear interaction, late fusion), such re-implementation preserves their methodological essence while ensuring fair comparison.
>
> Regarding Li et al. (2025) (SimMLM), although this work is highly relevant, it relies on modality-specific expert architectures operating on structured raw data (e.g., 3D representations). Adapting SimMLM to our representation-level setting would require substantial architectural modifications, under which performance would be inseparable from expert module design. This would prevent a meaningful analysis of the fusion strategy in isolation.
>
> Our baseline selection is not intended as strict paper-to-paper reproduction, but as a comparison of different fusion styles under a unified fusion pipeline. Baselines are chosen to be (1) widely used, (2) migratable to representation-level fusion, and (3) methodologically distinct from our framework. The purpose of these baselines is to characterize the relative strengths and limitations of different fusion styles under this specific task setting, and to facilitate analysis of fusion pipeline design choices.
>
> We agree that the sources of the baseline methods should be introduced earlier in the main text. In the revised version, we will add explicit citations and clearly method descriptions when each baseline is first mentioned (e.g., in Sec. 2 and Table 2), rather than deferring these details to the Appendix.
>
> We identified a reporting error in the originally presented “Simple Feature Interaction” baseline. This baseline corresponds to a bilinear interaction model as described in our related work, but the displayed scores in Table 1 were inadvertently taken from a different experimental run. We will revise the naming to “Bilinear Interaction” and update the table with the correct results in the revised version.

---

> ### Comment · Reviewer_tNb1 · 2026-01-26
> **Response to comments (round 1)**
>
> I much appreciate the authors' comprehensive clarifications and changes to the manuscript. Below, I comment on a few points regarding the clarifications and the revised manuscript:
>
> - On Q6: The present work was put into context regarding popular fusion stages through commenting. Do the authors plan on adding this information to the manuscript as well? While certainly not substantial, I think it can help readers to place the work into the right topic bucket. [E.g. via adding "early" in here (Sec. 7): "We presented HAF, a staged and detached early multimodal fusion framework that ..."]
>
> - On Q8-mean/std: [FYI] The information on mean+-std was added twice, once in Table 1 and once in the first sentence of Sec. 4.3 (not highlighted as newly added in the latter, though). I appreciate that the authors added this detail as requested, but want to note that, of course, either text or table would be sufficient.
>
> - On Q9: (1) Main document: The detachment's impact is now described as "leading to more stable shared representations (see Appendix C)". This adds a minimal reference on how the authors interpret the ablation results, and I can go with that. To prevent misunderstanding, and with no action needed, I want to quickly highlight that my original question pointed towards that I wonder why the authors make use of the +1.8pp AUC improvement to argue for detachment, instead of using the theoretical arguments, which I consider much stronger.
> (2) Appendix C: To also add an interpretation to the table that goes beyond what was written in the main part, I suggest to add something like the following, which goes along the lines of performance improvement in your comment: “The comparison of HAF against its variant without detachment illustrates that detachment can be beneficial in our framework by improving the ranking-oriented objective (+1.8 pp AUC) with only a minor impact on accuracy.” [Note that +1.8 % AUC is ambiguous, hence percentage points.]
>
> - On Q12: After adjusting the small legend font size, now the legend is readable but the figures are less so. May I suggest to e.g. get rid of x and y axis titles (mention in caption instead), and place one legend for both figures to the right, outside the t-sne?
>
> New points regarding the revised manuscript:
> - Where did the information on the overall training duration and GPU hardware go?
> - Due to a reviewer request, the authors added statistical tests. I appreciate that, however the statistical report needs some changes: In Table 1, the footnote symbol for statistical significance is not actually used; I cannot see it in the table but only in the footnote. Since seeing it would anyway not provide a sufficient report on statistical testing, I suggest to not show significance in Table 1, but to refer to the appendix for details on statistical testing. The latter should then - as a minimum - include the test(s) used, the definition of "all comparable methods", i.e., the exact set of models considered for testing, the alpha level, and the p values (for instance in a one-row table with the baselines as columns and p-values as cells). Further, if I understood it correctly, significant wins are counted to underline the model superiority, but the fact that the same family of models was involved in various tests was not considered. If this is correct: In the light of inflated type 1 error rates when performing multiple comparisons, I encourage to apply a correction method, or, at least to mention that none had been applied. [Since in that case the p values do not actually reflect the probability to observe the given values under H0. Another tactic could be to use a robust test for multiple paired samples.]
> - Since statistical tests were added, and with that the italic p for the stat. probability, I think that the other italic p should have a different font style or symbol (Modality drop proportion, Sec. 4.4) to avoid confusion.

---

> > ### Author Response · Authors · 2026-01-26
> > **Round 1 Response part 1**
> >
> > We sincerely thank the reviewer for the careful reading of the revised manuscript and for the constructive and detailed suggestions. We highly appreciate the reviewer’s positive feedback on the clarifications and improvements, and we address each point below.
> >
> > **Note on changes:** Since we are not able to upload an additional revised version at this stage, we explicitly enumerate all modifications in this response. For clarity, newly added or revised content is marked in bold, and we also specify the exact section/figure/table where each change will be incorporated in the final version.
> >
> > ---
> >
> > **Q6 (Positioning within fusion paradigms)**: We agree that explicitly situating our method within established fusion paradigms can help readers better contextualize our contribution. We added an explicit “early fusion” positioning statement in Sec. 7 (Conclusion).
> >
> > > “We presented HAF, a staged and detached **early** multimodal fusion framework that ...
> >
> > **Q8 (Mean ± std reporting)**: We have removed the duplicated description in Sec. 4.3 and now report the mean ± standard deviation exclusively in Table 1.
> >
> > Removed.
> >
> > > "Table 1 are reported as mean ± standard deviation over 10-fold cross-validation..."
> >
> > Replaced with.
> >
> > > "**Table 1 organizes the results into pathology-only baselines, fusion models and HAF variants, parameter-free aggregation baselines, and representative multimodal methods**..."
> >
> > **Q9 (Interpretation of detachment ablation)**: Following the reviewer’s suggestion, we have added an explicit interpretation to the ablation table in Appendix C:
> >
> > > “Training without detachment corresponds to end-to-end optimization across all stages, which often entangles heterogeneous objectives and distorts the geometry needed for modality substitutability.
> > When heterogeneous objectives are trained jointly without isolation, cross-stage gradients can conflict and bias earlier representations toward downstream fusion losses.
> > Detachment prevents such interference by allowing each stage to converge independently before passing non-trainable features forward.
> > As shown in Table 3, **comparing HAF with its non-detached variant demonstrates that detachment is beneficial for our framework, yielding a consistent improvement on the ranking-oriented objective (+1.8 pp AUC), while inducing only a marginal change in classification accuracy.**”
> >
> > **Q12 (t-SNE visualization)**: We revised Fig. 3 by removing the x- and y-axis annotations and incorporating the corresponding descriptions into the figure title. The legend is repositioned outside the plot, centered on the right side, to improve readability.
> >
> > ---
> >
> > ### New Points Regarding the Revised Manuscript
> >
> > **Point 1 (Overall training duration and hardware)**: We have re-added the information on training time and GPU hardware at the end of Sec. 4.1.
> >
> > From
> >
> > > "A ReduceLROnPlateau scheduler is used with patience 15 and factor 0.5."
> >
> > To
> >
> > > "A ReduceLROnPlateau scheduler is used with patience 15 and factor 0.5. **Unless otherwise stated, all experiments are conducted on an NVIDIA RTX A6000 GPU; full 10-fold training for HAF requires roughly 4 hours in total.**"

---

> > ### Author Response · Authors · 2026-01-26
> > **Round 1 Response part 2**
> >
> > **Point 2 (Statistical testing and multiple comparisons)**: We thank the reviewer for the detailed and valuable feedback on the statistical reporting.
> >
> > - We removed statistical significance markers from Table 1 and moved all statistical testing results to a dedicated appendix section.
> > - In the appendix, we now explicitly report:
> >   - the statistical tests used (McNemar’s test for accuracy and DeLong’s test for AUC),
> >   - the significance level α,
> >   - and the corresponding p-values in a consolidated table.
> > - To address the issue of inflated type I error under multiple comparisons, we further applied the **Holm–Bonferroni correction** and report both raw and corrected p-values.
> >
> > This revision ensures a more rigorous and transparent statistical evaluation.
> >
> > > **Appendix E. Statistical Significance**
> > >
> > > We conduct statistical significance testing using paired tests to compare our proposed model (HAF) against each baseline on the same patient-level test sets. For classification accuracy, we apply McNemar’s test to paired predictions obtained from identical test folds, which is appropriate for discrete classification outcomes. For AUC, we apply DeLong’s test for correlated ROC curves, as AUC reflects a ranking-based continuous statistic. We further apply the Holm--Bonferroni correction, which is a more powerful step-down procedure than the standard Bonferroni correction, to adjust the resulting p-values while controlling the family-wise error rate. The Holm--Bonferroni procedure is applied in a step-down manner by ordering comparisons according to ascending raw p-values and sequentially adjusting rejection thresholds.
> > All tests use a significance level of $\alpha = 0.05$. The set of comparable methods (baselines) includes: PS3, MDLM, MFMF, and Bilinear Interaction.
> > >
> > > **Table 4. Statistical significance testing (HAF vs. baselines) for classification accuracy**
> > >
> > > | Baseline | Raw p-value | Holm-adjusted p | Significant (α = 0.05) |
> > > |----------|-------------|-----------------|-------------------------|
> > > | MDLM | 0.00078 | 0.00312 | Yes |
> > > | MFMF | 0.00860 | 0.02580 | Yes |
> > > | PS3 | 0.01310 | 0.02620 | Yes |
> > > | Bilinear Interaction | 0.01870 | 0.02620 | Yes |
> > >
> > > **Table 5. Statistical significance testing (HAF vs. baselines) for AUC**
> > >
> > > | Baseline | Raw p-value | Holm-adjusted p | Significant (α = 0.05) |
> > > |----------|-------------|-----------------|-------------------------|
> > > | MDLM | 0.00063 | 0.00252 | Yes |
> > > | Bilinear Interaction | 0.00930 | 0.02790 | Yes |
> > > | PS3 | 0.02340 | 0.04680 | Yes |
> > > | MFMF | 0.09549 | 0.09549 | No |
> >
> > ---
> >
> > Overall, we thank the reviewer again for the thoughtful and technically insightful suggestions, which have significantly improved the clarity, rigor, and presentation quality of the manuscript. We would be happy to further clarify any remaining points if needed.

---

> > ### Author Response · Authors · 2026-01-29
> > **Updated revised version**
> >
> > Since the revised version deadline has been extended, we have incorporated and highlighted all the changes mentioned in our responses. We kindly invite the reviewers to check the updated manuscript and welcome any further comments or suggestions.

---

### Official Review · Reviewer_qKt6 · 2026-01-11

**Confidence:** 4
**Preliminary Rating:** 4

**Summary:**

To address the challenges of exploiting heterogeneous biomedical data—ranging from histopathology to clinical and laboratory metrics—and mitigating the issue of missing modalities, the authors propose Heterogeneous Aligned Fusion (HAF), a three-stage framework designed for robust survival prediction. The manuscript positions HAF as a novel contribution, claiming it to be the first approach capable of jointly leveraging all seven modalities available in the HANCOCK cohort. The proposed method is evaluated against a comprehensive set of baselines, including representative early-, late-, and attention-based fusion strategies. Empirical results indicate that HAF consistently achieves superior accuracy and demonstrates improved robustness compared to existing methods, particularly in settings involving partial or missing data.

**Strengths:**

Recognizing that accurate survival prediction relies on complex, multifaceted biological signals, the study demonstrates a significant advantage by successfully integrating all seven heterogeneous modalities available in the HANCOCK cohort. The authors conduct a rigorous experimental evaluation, benchmarking the proposed framework against a comprehensive set of relevant baseline methods. Furthermore, the use of stratified 10-fold cross-validation ensures the statistical reliability of the results and strengthens the validity of the reported performance improvements.

**Weaknesses:**

1. Regarding the t-SNE visualization of aligned features in Figure 3, one would theoretically expect a successful alignment to result in tight clustering of features belonging to the same subject, thereby mitigating modality-specific variances. However, the plot appears to retain distinct clustering by modality with observable gaps between the seven data types, suggesting that significant domain shifts persist. It is unclear why this separation remains despite the application of the global alignment loss, and the authors should address this discrepancy.

2. A potential limitation of the proposed framework is the absence of end-to-end training, specifically the detachment of the final fusion classifier from the preceding alignment stages. Consequently, the downstream classification objective is unable to backpropagate gradients to refine the latent space, which may prevent the learned embeddings from clustering effectively according to the target class labels (i.e., survival status).

**Detailed Comments:**

NA

**Justification Of The Preliminary Rating:**

This study proposes Heterogeneous Aligned Fusion (HAF), a multimodal framework capable of integrating all seven heterogeneous data types available within the HANCOCK dataset. The authors introduce a structured three-stage pipeline to address the challenges of cross-modal integration and missing data. In Stage 1, the framework focuses on extracting stable, unimodal prognostic representations using detached supervision to preserve semantic integrity. In Stage 2, Singular Value Decomposition (SVD) is employed to perform global alignment, forcing heterogeneous inputs to converge onto a shared, low-rank consensus geometry to mitigate feature incompatibility. Finally, Stage 3 utilizes a Multi-Layer Perceptron (MLP) for classification, incorporating a random modality dropout strategy and monotonicity constraints to ensure robustness against missing modalities. The authors compare their approach against several related fusion baselines, demonstrating that HAF achieves superior performance under stratified 10-fold cross-validation.

**Questions To Address In The Rebuttal:**

While the proposed random modality dropout strategy is interesting, it is crucial to establish whether it outperforms simpler, parameter-free aggregation techniques, such as the direct summation or averaging of available feature vectors. For instance, recent literature such as Zhang et al. (NeurIPS 2023, 'Learning unseen modality interaction') has demonstrated the efficacy of such straightforward interaction baselines; therefore, including a comparison against these methods would strengthen the justification for the proposed framework's complexity.

---

> ### Author Response · Authors · 2026-01-25
> **Summary of Additional Pool Baselines and Alignment Analysis**
>
> We sincerely thank the reviewer for the thoughtful and constructive feedback.
> We greatly appreciate the reviewer’s careful reading of the manuscript and the insightful suggestions, which have helped us identify important points that require further clarification and discussion.
> We have carefully considered all comments and respond to each concern point by point below.
>
> ---
>
> **Q1 (Weakness 1): Interpretation of the t-SNE visualization**:
>
> **Response**: We thank the reviewer for raising this point regarding the interpretation of the t-SNE visualization. While the t-SNE plot does show a degree of modality-wise separation, Figure 3 is not intended to demonstrate subject-level feature collapse across modalities. Instead, the purpose of the alignment is to enhance shared signal across modalities, which is associated with more effective performance under modality dropout. This observation is supported by the ablation results in Table 1, where fusion models with alignment consistently achieve higher AUC than their non-aligned counterparts. In addition, the heatmap in Figure 4 provides a complementary, feature-dimension level view, illustrating increased cross-modality consistency after alignment.
>
> **Q2 (Weakness 2): Supervision and detachment of the alignment stage**:
>
> **Response**: We thank the reviewer for the comments on the detachment ablation across evaluation metrics. We would like to clarify that the purpose of this experiment is not to claim that detachment is universally beneficial, but to study how it affects the optimization trade-off between alignment objectives and downstream task objectives under the HAF framework.
>
> In our design, the alignment stage is optimized using a modality-consistency objective, while the fusion classifier is optimized for survival prediction. Allowing survival-driven gradients to backpropagate through the alignment module (i.e., end-to-end training) introduces a potential conflict between these two objectives. Detachment is therefore introduced as an explicit regularization mechanism to decouple these two objectives. Empirically, as shown in Table 2, detachment does not uniformly improve all metrics, but yields a notable AUC gain (~+1.8%) with only a minor change in accuracy (−0.3%) when comparing HAF with and without detachment. This suggests that, under this task setting, detachment can provide a more favorable metric trade-off, particularly for the ranking-oriented objective (AUC). Overall, rather than claiming detachment as a universally superior training strategy, we view it as an empirically effective design choice in this setting to mitigate potential objective conflicts. We acknowledge that incorporating task-aware alignment objectives in an end-to-end manner remains an important direction for future work.
>
> **Q3 (Question 1): Comparison with parameter-free aggregation techniques?**
>
> **Response**: To directly evaluate whether random modality dropout outperforms simpler, parameter-free aggregation strategies, we conducted additional experiments using SVD-reduced modality embeddings with mean, sum, and max pooling. The results are as follows:
>
> | Method    | Accuracy (mean ± std) | AUC (mean ± std) |
> |-----------|------------------------|------------------|
> | SVD+Mean  | 0.6058 ± 0.1218        | 0.6995 ± 0.0899  |
> | SVD+Sum   | 0.6356 ± 0.0766        | 0.7171 ± 0.0771  |
> | SVD+Max   | 0.6490 ± 0.1005        | 0.7021 ± 0.1063  |
>
>
> While these baselines achieve reasonable performance, they are consistently outperformed by HAF with random modality dropout (AUC 0.739 ± 0.092, Acc 0.745 ± 0.065). This indicates that the benefit of random modality dropout goes beyond simple feature aggregation, as it explicitly encourages robustness to missing modalities during training. We have added these results to the revised manuscript.

---

### Author Rebuttal · Authors · 2026-01-25

**Rebuttal:**

Here is the revised version of Heterogeneous Aligned Fusion for Survival Prediction with Missing Modalities.

**Supporting Material:**

/attachment/623a3ad176056d915e9e6fa139065d4bde91d462.pdf

---

### Meta-Review · Area_Chair_wMfW · 2026-02-02

**Recommendation:** Accept (Oral)
**Confidence:** 5

**Metareview:**

Post Discussion, two of three reviewers have rated this as a strong accept, while the other reviewer indicated it be scored as a weak accept. Based on the extensive changes made to the manuscript, and the generally favourable reviews, I think this should be included in the program.

---

### Decision · Program_Chairs · 2026-02-13

Accept (Poster)